

# Tubular epithelial progenitors are excreted in urine during recovery from severe acute kidney injury and are able to expand and differentiate *in vitro*

Daniela Gerges[1], Zsofia Hevesi[2], Sophie H. Schmidt[1], Sebastian Kapps[1], Sahra Pajenda[1], Barbara Geist[3], Alice Schmidt[1], Ludwig Wagner[1] and Wolfgang Winnicki[1]

[1] Division of Nephrology and Dialysis, Department of Medicine III, Medical University Vienna, Vienna, Austria
[2] Center for Brain Research, Medical University Vienna, Vienna, Austria
[3] Department of Biochemical Imaging and Image-guided Therapy, Division of Nuclear Medicine, Medical University Vienna, Vienna, Austria

Corresponding author
Daniela Gerges,
daniela.gerges@meduniwien.ac.at

## ABSTRACT

**Background:** Acute kidney injury (AKI) is a serious condition associated with chronic kidney disease, dialysis requirement and a high risk of death. However, there are specialized repair mechanisms for the nephron, and migrated committed progenitor cells are the key players. Previous work has described a positive association between renal recovery and the excretion of tubular progenitor cells in the urine of kidney transplant recipients. The aim of this work was to describe such structures in non-transplanted AKI patients and to focus on their differentiation.

**Methods:** Morning urine was obtained from four patients with AKI stage 3 and need for RRT on a consecutive basis. Urine sediment gene expression was performed to assess which part of the tubular or glomerular segment was affected by injury, along with measurement of neprilysin. Urine output and sediment morphology were monitored, viable hyperplastic tubular epithelial clusters were isolated and characterized by antibody or cultured *in vitro*. These cells were monitored by phase contrast microscopy, gene, and protein expression over 9 days by qPCR and confocal immunofluorescence. Furthermore, UMOD secretion into the supernatant was quantitatively measured.

**Results:** Urinary neprilysin decreased rapidly with increasing urinary volume in ischemic, toxic, nephritic, and infection-associated AKI, whereas the decrease in sCr required at least 2 weeks. While urine output increased, dead cells were present in the sediment along with debris followed by hyperplastic agglomerates. Monitoring of urine sediment for tubular cell-specific gene transcript levels NPHS2 (podocyte), AQP1 and AQP6 (proximal tubule), and SLC12A1 (distal tubule) by qPCR revealed different components depending on the cause of AKI. Confocal immunofluorescence staining confirmed the presence of intact nephron-specific epithelial cells, some of which appeared in clusters expressing AQP1 and PAX8 and were 53% positive for the stem cell marker PROM1. Isolated tubule epithelial progenitor cells were grown *in vitro*, expanded, and reached confluence within 5–7 days, while the expression of AQP1 and UMOD increased, whereas PROM1 and Ki67 decreased. This was accompanied by a change in cell morphology from a disproportionately high

nuclear/cytoplasmic ratio at day 2–7 with mitotic figures. In contrast, an apoptotic morphology of approximately 30% was found at day 9 with the appearance of multinucleated cells that were associable with different regions of the nephron tubule by marker proteins. At the same time, UMOD was detected in the culture supernatant.

**Conclusion:** During renal recovery, a high replicatory potential of tubular epithelial progenitor cells is found in urine. *In vitro* expansion and gene expression show differentiation into tubular cells with marker proteins specific for different nephron regions.

## INTRODUCTION

Acute kidney injury (AKI) occurs due to various conditions such as sepsis, hypoxia, trauma, and exposure to toxins and affects 13.3 million people every year worldwide of which 1.7 million die (*Mehta et al., 2015*; *Peerapornratana et al., 2019*; *Vincent et al., 2006*). The most severe form of AKI—stage 3—is associated with the need for renal replacement therapy (RRT) in 23.5% of cases and can lead to persistent chronic kidney disease (CKD) (*Khwaja 2012*). However, during an episode of AKI, it is unpredictable whether a patient will recover or develop persistent kidney failure. Chronic kidney disease is not only associated with high socio-economic costs and need for RRT but also leads to a dramatic increase in mortality due to associated cardiovascular consequences, and only kidney transplantation (KTX) can provide improvement in this respect. Therefore, a major goal of AKI research should be to identify cellular factors associated with recovery and potentially use them as therapeutic targets in the future to prevent CKD and need for KTX.

Renal recovery relies upon a local renal epithelial progenitor pool and former studies have shown that appearance of renal epithelial progenitor clusters, called nephrospheres, in urine of kidney transplanted AKI (KTX-AKI) patients indicate recovery (*Knafl et al., 2019*). Therefore, there seems to be an urgent need to investigate whether nephrospheres can also be found in non-KTX-AKI patients and if so, whether their occurrence is also an indication of recovery in this patient population.

Repair mechanisms are extremely complicated and orchestrated as the biological function of each nephron segment is complex, relying upon the expression of highly specialized ion channels, ion sensing proteins, enzymes, and ciliary motor proteins. These proteins define into which segment of the nephron the cells have differentiated (*Abedini et al., 2021*; *Harari-Steinberg et al., 2020*). Therefore, bioengineering of nephrons seems almost impossible due to the detailed and sophisticated architecture of the organ (*Harari-Steinberg et al., 2020*; *Rahman et al., 2020*).

In the human fetus, nephron development relies on multipotent stem cells and nephron formation completes by the gestational week 34 (*Pleniceanu et al., 2018*). However, committed progenitors remain interspersed within the nephron epithelia in Bowman's
capsule and tubular segments until adulthood (*Romagnani & Remuzzi, 2013*). Renal recovery does not depend on bone marrow-derived stem cells (*Duffield et al., 2005*), as was assumed in the past, but rather on the replicative potential of these epithelial progenitor cells (*Humphreys et al., 2008*; *Romagnani & Remuzzi, 2013*). Thereby, after AKI, only renal progenitor cells proliferate and are responsible for cellular regeneration while other tubular cells endoreplicate and become polyploid leading to tubular cell hypertrophy (*Lazzeri et al., 2018*) in mouse models. It is estimated that both processes are required to ensure renal recovery. In this process, balance is key as exuberant hypertrophy leads to fibrosis and CKD, while enhancement of renal progenitor proliferation has been shown to induce clonal papillary adenoma and renal cell carcinoma (RCC) (*Peired et al., 2020*). This observation, that AKI may lead to papillary RCC, has also been made in two different human cohorts (*Peired et al., 2020*, *2021*). In accordance therewith, nephrogenic adenomas have been found in KTX patients and it has been shown that they are derived from tubular cells of the renal transplant.

Nevertheless, cellular repair is critical after AKI to prevent from the development of CKD. To achieve this, renal progenitors can proliferate and differentiate into different functional segments of the nephron. In this context, hyperplastic tubular epithelial cell clusters together with apoptotic cells derived from over-proliferation have been shown in a rat model of AKI *in vivo* (*Shimizu & Yamanaka, 1993*) and have also been found in human urine (*Nguyen & Smith, 2004*). As mentioned above, they can cause engraftments in the urinary bladder and initiate nephrogenic adenomas (*Mazal et al., 2002*). In an earlier study, our group was able to isolate hyperplastic clusters of renal progenitor cells from urine of KTX patients with AKI stage 3 and demonstrate that their appearance in urine showed a positive correlation with recovery from AKI, as 100% of all patients with nephrospheres in urine recovered, while only 39.3% without nephrosphere-excretion recovered from AKI (*Knafl et al., 2019*; *Shimizu & Yamanaka, 1993*). However, in our previous study, none of the non-transplanted AKI patients showed nephrosphere excretion in the urine because of minor grades of AKI. We therefore addressed the question of whether nephrosphere excretion could also be detected in non-KTX patients with severe AKI requiring RRT, as we hypothesized that nephrosphere excretion is related to severe renal damage.

In this study, renal progenitors were isolated from human morning urine of patients with recent AKI stage 3 and need for RRT. Kidney-specific gene expression using qPCR, morphological cell monitoring and measurement of urine and serum biomarkers were performed over the period of recovery. In parallel, a tissue culture of isolated tubular cells was established, and the *in vitro* proliferation and differentiation were monitored by protein immunostaining and gene expression.

## MATERIALS AND METHODS

### Patient enrollment and sample collection

Among 700 patients admitted to the Department of Nephrology presenting with stage III AKI, four patients with excretion of nephrospheres in urine were enrolled on a continual basis after giving oral and written informed consent. The study was approved by the ethics

committee of the Medical University of Vienna under the number 1043/2016. Catheter urine was obtained every morning from the designated port and was immediately processed.

## Cell isolation and *in vitro* culture

Morning urine was centrifuged in 50 ml conical tubes at 1,500 RPM in order to pellet cells. The resultant cell pellet was suspended in tissue culture medium (RPMI 1640 supplemented with 10% fetal calf serum) which was layered onto Ficoll-Paque Plus (GE Healthcare, Chicago, IL, USA) and centrifuged at 2,000 RPM for 20 min. The interface was taken off and washed using tissue culture medium. The cell pellet was then resuspended and plated into 12-well plates. The second day the culture medium was replaced with tubular cell culture medium. This type of medium was prepared immediately before cell feeding by combining the tissue culture medium with (30%) ProxUp RPTEC Growth Medium (Evercyte, Vienna Austria). Cell feeding was carried out every second day, thereby dead cells were washed off and taken from the well together with old tissue culture medium.

## Preparation of cytoslides

Epithelial cells monolayer was washed twice with PBS and then incubated with trypsin/EDTA over 3 min. Trypsin was then inactivated by tissue culture medium and suspended cells were washed with tissue culture medium. A total of 100 μl of cell suspension was applied into the cytocentrifuge funnel which was spun at 1,200 RPM for 3 min. The resultant cytopreparation was air-dried and either immediately stained using a routine H/E staining for visualization of cell density or frozen at −20 °C wrapped in aluminum foil.

## Immunofluorescence staining

Cytopreparations of *ex vivo* urinary cells and of cultured epithelial cells were fixed in acetone for 5 min. A water repellent barrier was drawn around the area where cells had been placed by the cytocentrifuge using a PAP Pen (SCI, science service, Munich, Germany). Following wetting of the cell containing area with 20 μl PBS the antibodies were applied. CD133/1 (Prominin 1), AQP1 (aquaporin 1), PAX8 a kidney specific transcription factor (*Kaminski et al., 2016*), CD10 (neprilysin). The dilution is described below. Incubation was carried out at room temperature for 2–3 h under constant shaking in a moist chamber. Before application of the secondary antibodies, slides were washed in PBS for 10 min. As secondary antibodies, goat anti-rabbit Alexa Fluor 488 (diluted 1:400) and Rhodamine (TRIC)-conjugated AffiniPur F(ab)2 goat anti-mouse (diluted 1:200) were applied and incubated for 1 h under constant shaking at room temperature. As nuclear staining, DAPI/PBS was incubated for 5 min before final washing for 10 min twice. Slides were mounted in Vectashield mounting medium for immunofluorescence (Vecotor Laboratories, Burlingam, CA, USA) and covered with a coverslip. A Zeiss Axiovert confocal microscope was used for picture recording. Pictures were further processed using Photoshop Version 6.

## RNA isolation

The pellet of 7–10 ml morning urine was lysed in 1,000 μl of Trizol and kept for 10 min at room temperature and frozen at −20 °C until RNA isolation. Before addition of 200 μl chloroform the Trizol lysate was thawed. The content was then vigorously shaken and spun at 12,000$g$ for 10 min for phase separation. The aqueous phase was pipetted into a separate tube and mixed with 500 μl isopropanol. Following an incubation period of 10 min at room temperature tubes were spun at 12,000$g$ for 10 min at 4 °C. The RNA pellet was washed with 500 μl 75% ethanol, and the pellet was shortly dried and finally dissolved in about 20 μl RNAse free water. The RNA concentration was evaluated at a Nano drop device.

## Reverse transcription and quantitative PCR

Eight hundred ng of RNA was combined with dNTPs and random primers and incubated for 5 min at 65 °C followed by rapid chilling on ice-water. First stand buffer, DTT, and RNaseOUT and Superscript III was added before increasing incubation temperature to 42 °C for cDNA synthesis for 50 min. Enzyme activity was stopped by heating the sample at 70 °C for 10 min. The resultant cDNA concentration was diluted up to 80 μl and out of this 2 μl were added to the qPCR setup mix. This consisted of TaqMan mastermix and gene specific probe sets. Each sample was set up in duplicate in a final volume of 10 μl.

## UMOD ELISA from tissue culture

The human uromodulin ELISA was purchased from BioVendor (RD191163200R) and performed as described in the supplied test manual. In brief, tissue culture fluid was applied to the test well accompanied with a standard series as supplied in the test kit. Following 60 min of incubation at room temperature under constant shaking the plate was washed three times with wash buffer. Following this, the diluted biotinylated detection antibody was applied to each well and again incubated for 60 min at room temperature. Following the same washing procedure as above, the streptavidine/HRP was applied to each well and reacted for 30 min. After a final washing step, the ELISA was developed with TMB substrate incubating for 10 min under light protection. The reaction was stopped using the stop solution and read with an ELISA reader. Concentrations were calculated according to the standard curve.

## Neprilysin ELISA using urine

The human neprilysin ELISA was purchased from RayBiotech and performed according to the description in the test manual. In brief, frozen urine was thawed mixed and centrifuged at 3,000 RPM before loading. A total of 20 μl of assay diluent were dispensed in each well before loading 80 μl of urine. Samples from time points with disease maximum had to be diluted up to 1:5 in provided assay diluent. Following 2 h incubation at room temperature under constant shaking the plate was washed three times with the provided wash buffer. This was followed by incubation with the biotinylated antibody using a dilution of 1:100 in assay diluent. Following an incubation period of 60 min and washing the plate three times on the plate washer, the streptavidine/HRP conjugate was dispensed into each well at the

recommended dilution of 1:300. Following an incubation period of 45 min and washing of the plate (three times) on the plate washer the antibody reaction was developed with TMB at room temperature under light protection. Following the application of 50 µl stop solution the reaction intensity was measured on the ELISA reader and neprilysin concentrations in samples were calculated according to the included standard curve.

## RESULTS

### Epithelial cell cluster excretion started with an increase in urine output

Among 700 screened patients, four non-KTX AKI stage III patients exhibited nephrosphere excretion in urine and were included into this study. The patient with ischemic AKI was a 34-year-old male, who exhibited peak serum creatinine (sCr)-levels of 13.57 mg/dL (Fig. 1A). The patients with toxic AKI was a 19-year old male with peak sCr-levels of 9.87 mg/dL; the patient admitted with nephritic AKI was a 56-year old male with peak sCr-levels of 11.91 mg/dL, and the patient with infection-associated AKI was a 29-year-old male with a peak sCr of 12.39 mg/dL (Figs. 1C, 1E, and 1G). All patients required RRT during recovery from AKI.

The occurrence of epithelial cell clusters in the urine started with increasing urine volume and represented a patient specific variable (day 2 to day 14) (Table 1, Figs. S1 and S2).

### The proximal tubular epithelial cell marker protein neprilysin and nephron derived epithelial cell clusters are found in urine of AKI stage three patients during recovery

On the first day that patients resumed urine production after anuria, urine collection was performed with morphological examination of urine sediment and measurement of gene expression. Morphological follow-up pictures of the urinary sediment are shown from a patient with toxic AKI in Fig. S1 and one with infection-associated AKI in Fig. S2. During the initial phase of renal recovery, patients with AKI of different etiologies: ischemic (Fig. 1A), toxic (Fig. 1C), nephritic (Fig. 1E), and infectious AKI (Fig. 1G) experienced a steady increase in urine output accompanied by decreasing urinary excretion of the cell injury marker neprilysin. The decrease in sCr showed a delay of about 5 days compared with urinary neprilysin as marker of tubular cell damage (*Bernardi et al., 2021*; *Knafl et al., 2017*; *Pajenda, Mechtler & Wagner, 2017*). Peak urinary neprilysin levels were significantly higher in infection-associated AKI (Fig. 1G). Renal recovery in terms of sCr levels and time was comparable among all four types of AKI.

In addition, RNA was extracted from 10ml of urine sediment, reverse transcribed and analyzed for expression of renal epithelial genes. The transcriptome of the tubular epithelial- and podocyte-specific genes, AQP1, AQP6, NPHS2, SCL12A1, was examined over a period of 11 days for toxic and ischemic and of up to 24 days for nephritic and infection-associated AKI. Expression of the podocyte marker gene podocin (NPHS2) varied but was highest on the first day of observation in patients with ischemic AKI (Fig. 1B) and increased by the fifth day in those with toxic AKI (Fig. 1D). Of note, no podocin was detected in the sediment of patients with nephritic AKI (Fig. 1F) and

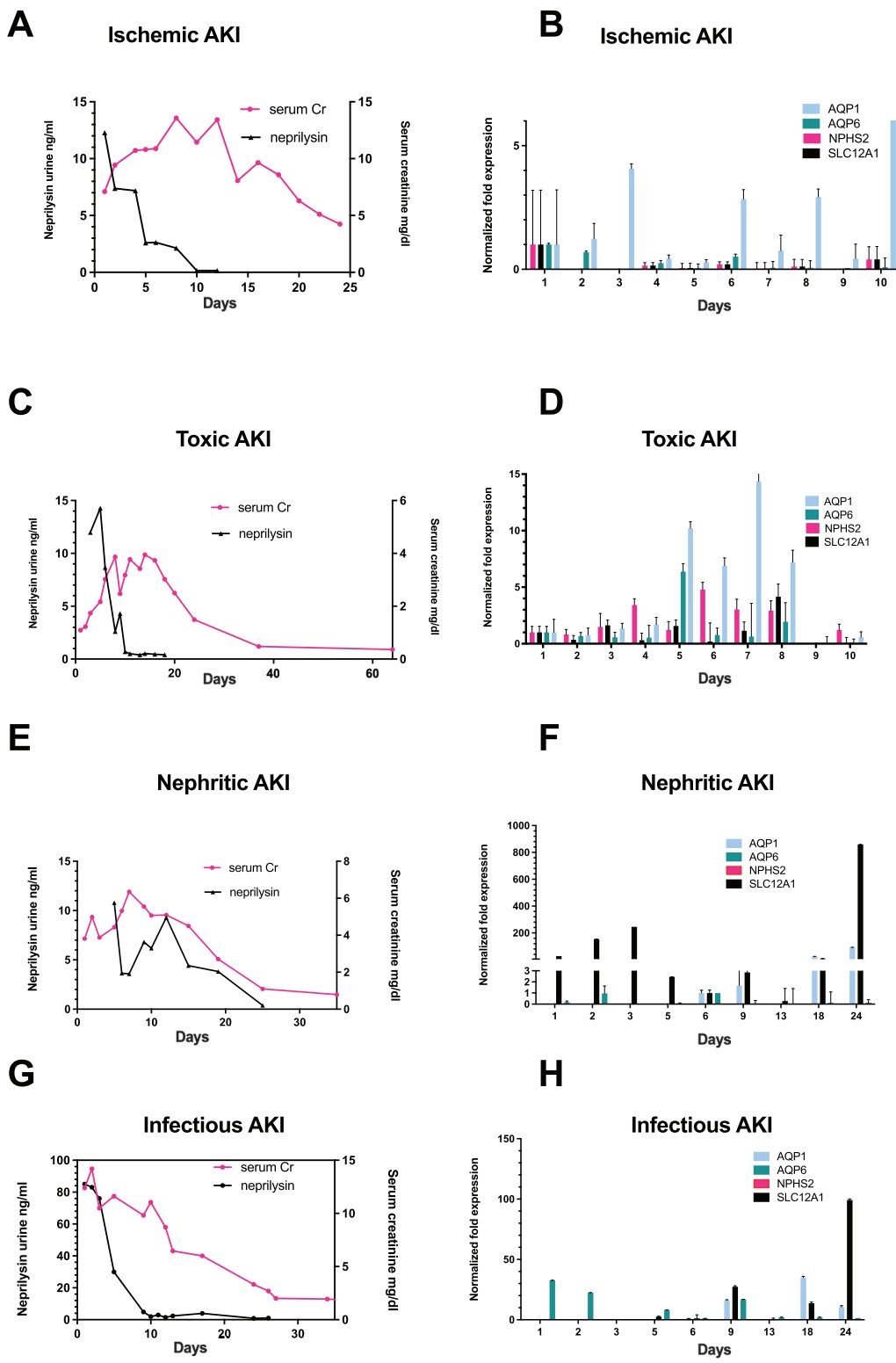

**Figure 1 Correlation of urinary neprilysin- and sCr-levels, and urinary expression of AQP1, AQP6, NPHS2 and SLC12A1 during recovery from AKI of different etiologies.** Urinary neprilysin, sCr and urinary expression of AQP1, AQP6, NPHS2, and SLC12A1 were measured during recovery from ischemic (A, B), toxic (C, D), nephritic (E, F) and infection-associated (G, H) AKI stage 3. The sCr-levels

**Figure 1 (continued)**
were monitored, and fluctuations within the first 13 days were attributable to the use of RRT. Urinary sediment cell and cell fragment gene expression was measured by qPCR. The first day of urine excretion was arbitrarily taken as the reference value (1st day in B and D, 6th day in F and H). All other days were compared with this value. Data are given as means ± SD.

**Table 1 Patient characteristics.** Patient characteristics with AKI etiology, peak-sCr-levels, day of peak-sCr, first day of sCr below 2.5 mg/dL, first day of nephrosphere appearance in urine and day of discharge.

| Etiology of AKI | Age (years) | Peak-sCr (mg/dL) | Day of peak sCr | Day of sCr < 2.5 mg/dL | 1st Day of nephrosphere | Day of discharge |
|---|---|---|---|---|---|---|
| Ischemic AKI | 32 | 13.57 | 8 | 29 | 10 | 21 |
| Toxic AKI | 19 | 9.87 | 14 | 28 | 2 | 28 |
| Nephritic AKI | 56 | 11.91 | 7 | 24 | 12 | 19 |
| Infectious AKI | 29 | 12.39 | 1 | 27 | 14 | 25 |

**Note:**
sCr = serum creatinine.

infection-associated AKI (Fig. 1H). Expression of the proximal tubule marker AQP1 increased in all four variants of AKI, toxic, ischemic, nephritic, and infection-associated, reflecting increasing numbers of hyperplastic epithelial cells in the urine (Figs. 1B, 1D, 1F and 1H). A similar pattern was found for the distal tubule cell-specific marker gene SLC12A1. Peak detection was found in patients with nephritic AKI and infection-associated AKI (Figs. 1F and 1H) at day 24, whereas in toxic AKI, peak levels were found during the early phase of recovery.

## Naive epithelial cell clusters express the kidney-specific transcription factor PAX8 and the stem cell marker PROM1 (CD133) in confocal immunofluorescence staining

In a first attempt of investigating the origin of urinary sediment cells, naive isolated cells from urine were examined. Immunofluorescence and confocal microscopy were performed to confirm the gene expression pattern and to evaluate the morphology of the excreted cells contributing to the expression landscape shown by qPCR (Figs. 1B, 1D, 1F and 1H). Most of the cell clusters examined expressed AQP1, and a subpopulation was positive for the stem cell marker PROM1 (CD133) (Fig. 2). All of cells visualized in cluster formation showed positive staining for PAX8, a kidney-specific transcription factor (Fig. 2). In the patient with ischemic AKI, no podocin-expressing cells could be detected.

## Isolated nephron derived epithelial cells from urine of patients recovering from stage 3 AKI proliferate in *in vitro* cell culture for a minimum of 7 days

Viable epithelial cells consisting of hyperplastic cell clusters (Fig. 3A) and individual cells were isolated by density gradient sedimentation of urine sediment cells and incubated in culture medium as described in the methods section. A rapid expansion from proliferating foci and migration over the entire free tissue culture plate was observed over the initial 5 days (Fig. 3B). As demonstrated in Fig. 3C a minor number of cells already

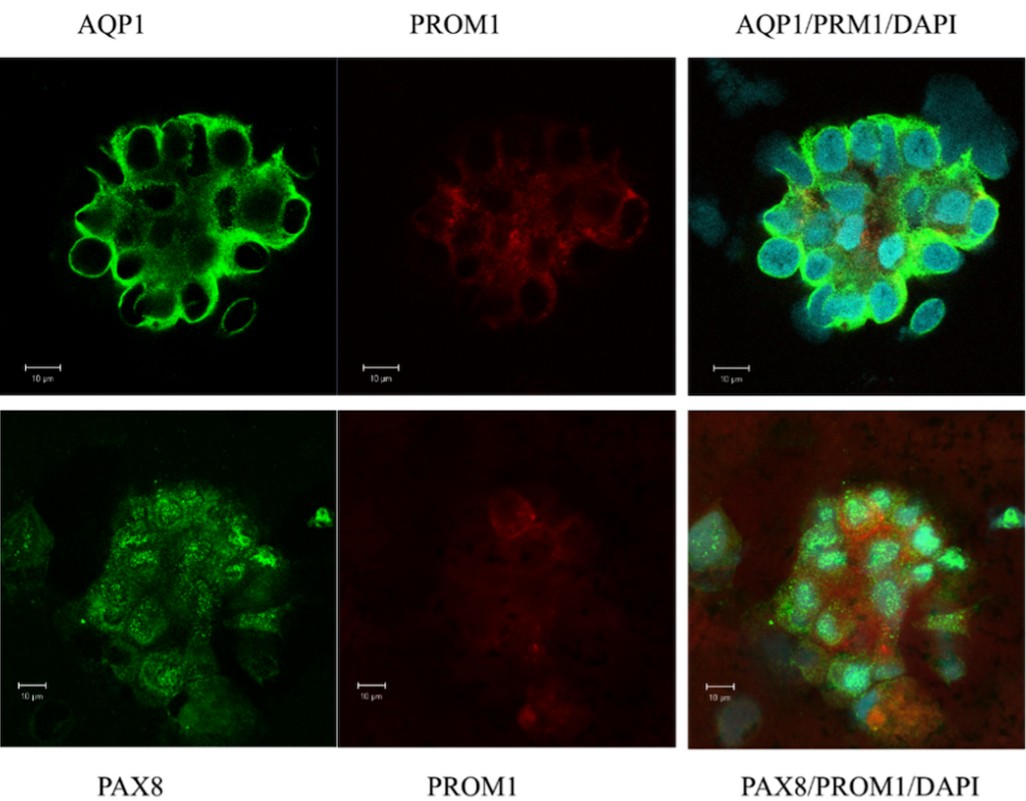

**Figure 2 Confocal immunofluorescence of clusters of renal epithelial cells excreted in urine during the phase of renal recovery.** AQP1 staining (green, upper panel) indicated derivation from proximal tubular origin. Positive PROM1 (red) directed towards progenitors. PAX8 (green, lower panel), a kidney-specific transcription factor, demonstrated the nephronic origin of the cells.

started to show apoptotic morphology with condensed chromatin and apoptotic bodies on day 7. Fifty percent of the cell population continued to exhibit disproportionate nuclear/cytoplasmic ratios, indicating mitotic and DNA synthesis stages of the cell cycle. This scenario changed with day 8–9 when 70% of adherent growing cells started to round up and detach from the culture plate.

### *In vitro* grown epithelial cells express genes affiliated with podocytes, the proximal and distal tubule, and the loop of Henle

In order to further define the *in vitro* growth and differentiation of tubular epithelial cells, total RNA of *in vitro* cultured cells was reverse transcribed, and the resultant cDNA was investigated by qPCR using gene specific probe sets indicative for defined nephron segments such as for proximal tubular epithelial cells (AQP1), the loop of Henle (SLC12A1) and distal tubular epithelia (SLC12A3) and podocytes (NPHS2). This revealed remarkable AQP1 expression in addition to SCL12A1 and NPHS2 (Fig. 4). NPHS2 was detected from day 8 of culture.

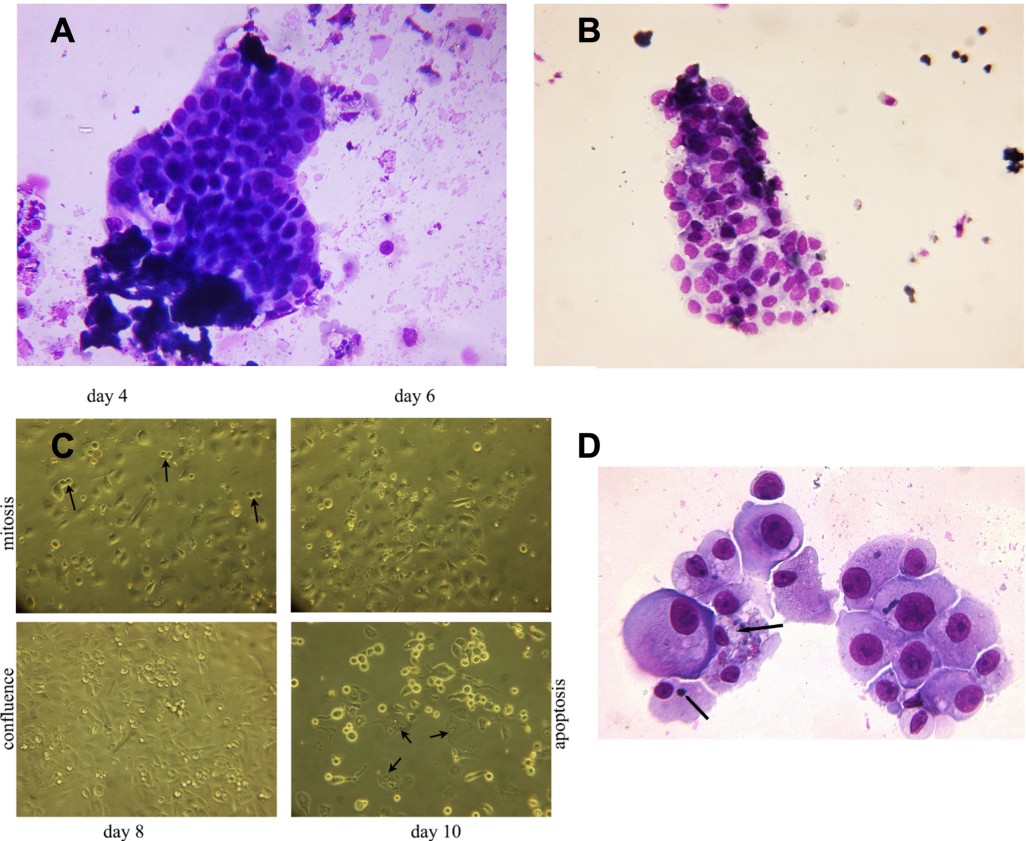

**Figure 3 Hyperplastic tubular epithelial cell clusters isolated from urine.** (A) Hyperplastic tubular cell agglomerates from a patient with ischemic AKI stage 3 including a cast. (B) Hyperplastic tubular epithelial cell cluster from a patient with infection-associated tubular necrosis. (C) Tubular epithelial cell clusters and progenitor cells isolated from urine by density gradient centrifugation grown in culture dishes in adherent mode. During mitosis in the phase of cytokinesis epithelial cells loosened from the dish and formed mitotic doublets (arrow). Mitotic cells were seen starting day 3 through day 8. This was followed by predominant apoptotic structure starting with day 9. Cells containing two or three nuclei (endocycling arrow) were observed. (D) H/E staining of cytopreparations of 8 day cultured human renal epithelial progenitor cells. Some of them appeared with disproportionate cytoplasmic nuclear size in mitosis, others possessed condensed nuclear chromatin and apoptotic bodies as typical indicators of apoptosis (arrow). Original magnification 400x.  

### *In vitro* grown cells exhibit PAX8, PROM1 and increasing levels of UMOD

As a next step, we thought it would be of interest to establish whether subpopulations of cells would differentiate into region-specific epithelia by translating specific marker genes into proteins. Therefore, immunofluorescence staining of *in vitro* cultured cells was performed.

   *In vitro* grown cells also showed positive staining for PAX8, and the protein distribution was spread throughout the cells, with more than 50% of cells also positive for PROM1 (Fig. 5). This changed rapidly over the time of *in vitro* culture. While the nuclear size decreased, the proportion of PROM1 positive cells minorized, the expression levels of AQP1 and of UMOD increased, reaching a maximum on day 9 (Fig. 5). When testing for

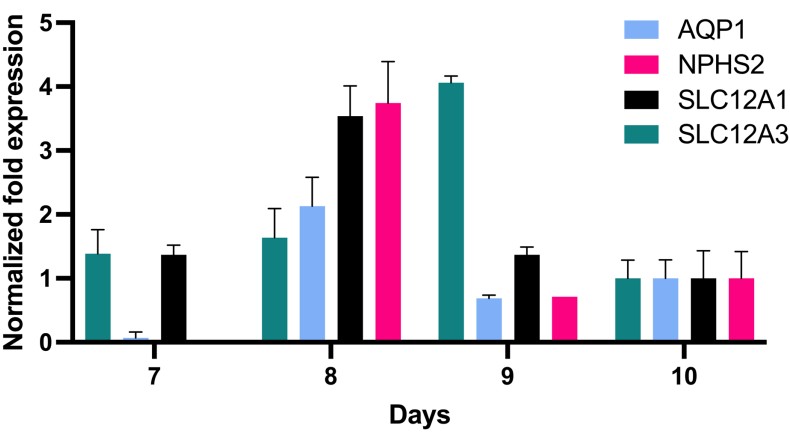

**Figure 4 Gene expression of epithelial cells *in vitro* grown for 7, 8, 9 and 10 days in tissue culture media.** Total RNA was isolated from cultured cells and qPCR was performed. Expression data of day 10 were arbitrarily chosen as reference and data from other days were compared to this. *In vitro* grown cells showed affiliation to podocytes (NPHS2) with a maximum on day 8 and to the proximal tubule (AQP1), the loop of Henle (SCL12A1) as well as the distal tubule (SLC12A3). Data are given as means ± SD.

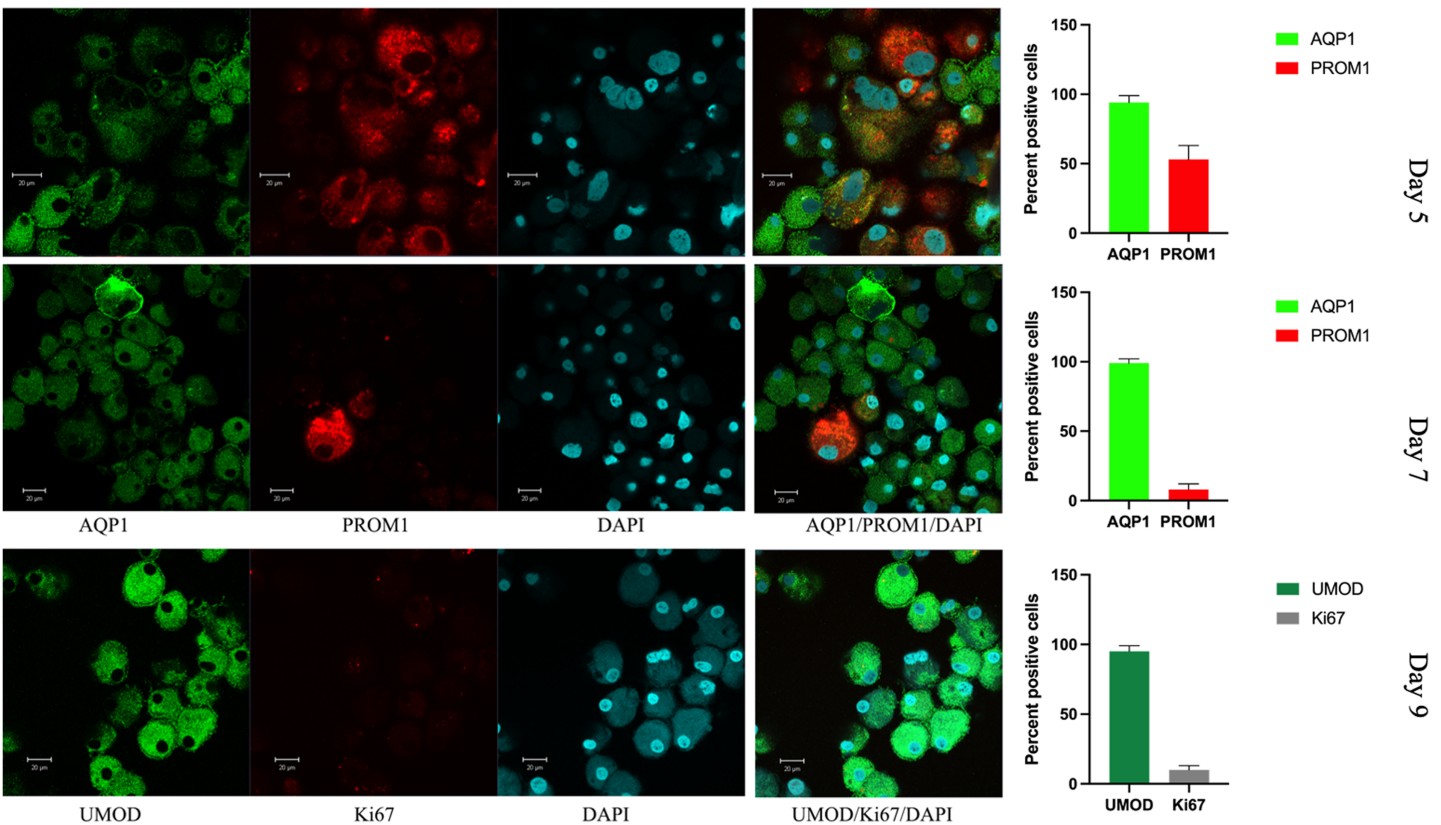

**Figure 5 Confocal immunofluorescence of renal epithelial cells 5, 7 and 9 days in culture.** Renal epithelial cells 5 days in culture stained positive for AQP1 (green) and PROM1 (red). Cells cultured for 7 days showed increased AQP1 staining (green) and reduced staining for PROM1 (red). Cells cultured for 9 days exhibited strong staining for UMOD (green), while Ki67 was much reduced (gray). Mean values for percent of positive cells and standard errors are given at the right for each day. This is a representative experiment out of three.

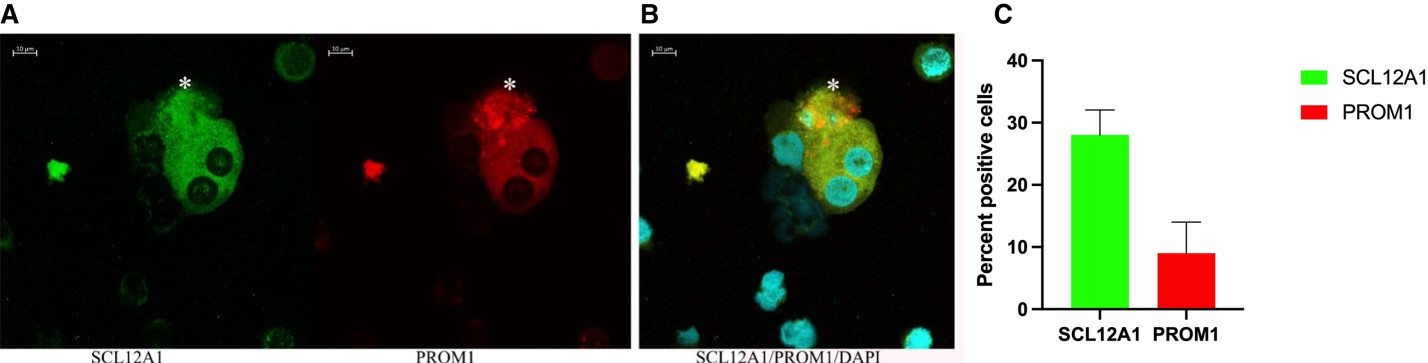

**Figure 6 Confocal immunofluorescence of renal epithelial cells 8 days in culture.** (A) Ten percent of SCL12A1 (green) positive cells also expressed PROM1 (red). (B) SLC12A1/PROM1/DAPI merged. The asterisk (*) represents an apoptotic SCL12A1-expressing cell containing apoptotic bodies. (C) Mean values for percent of positive cells and standard errors. This is a representative experiment out of two.

the proliferation marker Ki67 on day 5, 15% of cells stained positive (Fig. S3), the percentage of nuclear staining decreased and was almost absent on day 9, while tubular protein expression markers AQP1/UMOD increased (Fig. 5, bottom panel). Testing into different regions of the nephron the SLC12A1 expression was found in 28% of the outgrowing cells at day 8 of culture. Interestingly, these cells were also positive for PROM1 in 9%. Among these cells were 8% undergoing endocycling (Fig. 6). Measurement of UMOD in the culture supernatant increased starting from day 5 and reaching a maximum on days 8–9 (Fig. S3).

## DISCUSSION

Acute kidney injury represents a major risk for the development of CKD, need for dialysis and death. In the past, renal epithelial progenitor cells in urine have shown to be an excellent prognostic sign for recovery from AKI in kidney transplanted patients, however up to now, none of these cell agglomerates have been described in non-transplanted AKI patients (*Knafl et al., 2019*). Therefore, the presence of renal epithelial progenitor cells from urine sediment at baseline and during the period of renal recovery in AKI stage 3 was investigated and their differentiation into different nephronic segments was studied. The renal injury marker neprilysin in urine and the course of serum creatinine were monitored during this period. The proliferation potential of the isolated progenitor cells and their initial protein and gene expression were investigated and followed by *in vitro* propagation for 9 days until apoptosis. The *in vitro* production of UMOD was also detected in the tissue culture supernatant in this study.

There is no specific therapy for AKI in patient care to heal tubular injury and cure the microvascular damage. However, the tubule's capacity to self-repair is remarkable, especially in AKI stage 1 and 2 or at a young age (*Humphreys et al., 2008*). The mechanisms have been studied in mouse and rat models (*Romagnani, Rinkevich & Dekel, 2015*). As mentioned above, there are two types of cells found within the kidney tubules: renal progenitor cells, which proliferate and are responsible for cellular regeneration and cells that endoreplicate and generate hypertrophy (*Lazzeri et al., 2018*). Exuberant hypertrophy

appears to lead to fibrosis and CKD, while an overshooting progenitor proliferation leads to clonal papillary adenomas and RCCs (*Peired et al., 2020*, *2021*; *Venkatachalam et al., 2015*). It is therefore highly likely that the proliferative rate of epithelial progenitors could also be coupled with a certain rate of mutations coinciding with papillary RCC. When these tubular progenitors are incorporated into organoids, the overall growth and cell survival is much prolonged and even can give rise to nephrogenic structures when injected into SCID mouse kidney capsules (*Harari-Steinberg et al., 2020*). However, it seems inconceivable that this could lead to the formation of entire nephrons, although it has been a long-standing vision to create a self-organ from autologous mesenchymal stem cells (*Lang et al., 2013*; *Rahman et al., 2020*). Given the very long transplant waiting list of dialysis patients, various experiments have been performed with the injection of human mesenchymal stem cells into the nephrogenic site of embryonic rats and with the cultivation of ureteric buds in the omentum of uni-nephrectomized rats (*Yokoo et al., 2006*). Although there was progress, a clinically relevant step does not appear to be in the near future.

In this study, we were able to isolate viable progenitor cells from urine in the phase of recovery from AKI. Epithelial cell clusters were detected in urine from the first day of admission for AKI up to 24 days and their appearance seemed to be independent of the patient's age. However, due to the small sample size no clear statement can be provided.

In addition, the excretion rate of regenerative stem cells was highly variable among different individuals. Whether this depended on the renal age or related to the individual genetic background has to be worked out in future studies. It would be of great interest, whether this proliferative rate is of relevance for renal function recovery or determining the time span of a patients CKD degenerating to CKD5. Kidney organoids might support this type of research (*Bonventre, 2018*).

Urinary granulocyte and monocyte excretion has been demonstrated in renal recovery from ischemic injury and toxic tubular necrosis (Fig. S2). It is noteworthy that CSF-1 signaling was associated with tubular epithelial cell repair in mice (*Menke et al., 2009*). This suggests that this seemingly inflammatory process involving granulocyte diapedesis from juxtatubular capillaries is directly linked to the pathway of epithelial repair in renal tubules (*Goligorsky, 2008*). In addition, specific proteins—such as stanniocalcin-1—can inhibit exuberant reactive oxygen species production and thereby protect from extensive ischemic injury (*Huang et al., 2012*).

Moreover, the renal cell injury marker neprilysin in urine (*Bernardi et al., 2021*; *Pajenda, Mechtler & Wagner, 2017*) decreased rapidly after elimination of nephrotoxic substances or restoration of oxygenation. The course of urinary neprilysin showed that excretion of hyperplastic epithelial cells began with a short delay of approximately 2 days when urinary neprilysin had flattened.

Based on immunostaining and qPCR, it must be assumed that there is heterogeneity among the outgrowing tubular epithelial cells. Most likely, they do not reflect the *in vivo* conditions.

Single-cell RNA sequencing would have allowed determination of the stretch of the tubular part of the nephron from which single excreted urinary cells could have originated

(*Abedini et al., 2021*; *Menon et al., 2020*). This could not be performed in our laboratory setting, instead qPCR was performed from the entire urinary sediment, which represents a limitation of our study. Beyond that, the sample size of our study is small.

In our previous work, these hyperplastic epithelial progenitor clusters were only observed in KTX patients with ischemic reperfusion injury and were positive predictors of renal recovery (*Knafl et al., 2019*). Whether the occurrence of hyperplastic epithelial cell clusters in the urine of stage 3 non-transplanted AKI patients is associated with recovery from AKI can be suggested by this study, as all included patients recovered, but cannot be confirmed by our data, as no stage 3 non-transplanted AKI patients without excretion of epithelial cell clusters were included in this study. To our knowledge, this is the first study describing the occurrence of hyperplastic epithelial cell clusters in non-KTX patients recovering from AKI stage 3.

## CONCLUSIONS

The aim of this study was to investigate the presence of urinary tubular epithelial progenitor cells in patients during recovery from AKI stage 3 and to explore their growth behavior and differentiation *in vitro*. In the past, these structures have been shown to be strongly associated with recovery from AKI in KTX patients but could not be delineated in non-KTX patients (*Knafl et al., 2019*). This is the first study showing renal progenitor cell clusters in urine of non-transplanted patients with severe AKI stage 3 and RRT. These hyperplastic epithelial cell agglomerates expressed tubular marker proteins associated with podocytes (NSPH2), the proximal tubule (AQP1, AQP6), and the distal tubule (SLC12A1) as confirmed by qPCR. Confocal immunofluorescence staining confirmed that these epithelial cells were intact and nephron-specific, staining positive for AQP1 and PAX8. In addition, up to 90% of these cells expressed the stem cell marker PROM1, indicating their replicatory potential. Isolated in cell culture, these tubular epithelial progenitors grew, expanded, and reached confluence within 5 to 7 days, continuing to express AQP1 and UMOD, while the proliferation markers PROM1 and Ki67 decreased over time.

Single-cell RNA sequencing would have been beneficial, as it would have allowed determination of the section of the tubular part of the nephron from which single excreted urinary cells could be derived, but this was not possible in our setting. This represents a limitation of the study.

In conclusion, this is the first study describing hyperplastic tubular epithelial progenitor cell clusters in non-transplanted humans after AKI stage 3. Renal recovery appears to be reflected in high numbers of tubular cells with high replicative potential in the urine. Whether the occurrence of renal progenitor cell clusters in the urine of non-transplanted stage 3 AKI patients is also a positive predictor of recovery, as in KTX-AKI patients, and whether these clusters are associated with the occurrence of papillary adenomas and RCCs later in life, must remain an open question in this study and should be the subject of future investigations.

### Funding

This work was supported by the Medizinisch-Wissenschaftlichen Fonds des Bürgermeisters der Bundeshauptstadt Wien (No. 22100). The funders had no role in study design, data collection and analysis, decision to publish, or preparation of the manuscript.

### Grant Disclosures

The following grant information was disclosed by the authors:
Medizinisch-Wissenschaftlichen Fonds des Bürgermeisters der Bundeshauptstadt Wien: 22100.

### Competing Interests

The authors declare that they have no competing interests.

### Author Contributions

- Daniela Gerges conceived and designed the experiments, performed the experiments, analyzed the data, prepared figures and/or tables, authored or reviewed drafts of the article, and approved the final draft.
- Zsofia Hevesi analyzed the data, authored or reviewed drafts of the article, and approved the final draft.
- Sophie H. Schmidt performed the experiments, analyzed the data, prepared figures and/or tables, and approved the final draft.
- Sebastian Kapps performed the experiments, analyzed the data, prepared figures and/or tables, and approved the final draft.
- Sahra Pajenda conceived and designed the experiments, authored or reviewed drafts of the article, and approved the final draft.
- Barbara Geist conceived and designed the experiments, authored or reviewed drafts of the article, and approved the final draft.
- Alice Schmidt conceived and designed the experiments, authored or reviewed drafts of the article, and approved the final draft.
- Ludwig Wagner conceived and designed the experiments, performed the experiments, analyzed the data, prepared figures and/or tables, authored or reviewed drafts of the article, and approved the final draft.
- Wolfgang Winnicki conceived and designed the experiments, analyzed the data, prepared figures and/or tables, authored or reviewed drafts of the article, and approved the final draft.

### Ethics

The following information was supplied relating to ethical approvals (*i.e.*, approving body and any reference numbers):

This study was approved by the Ethics Committee of the Medical University of Vienna (1043/2016).

## Data Availability

The raw data is available in the Supplemental Files.

## Supplemental Information

Supplemental information for this article can be found online at http://dx.doi.org/10.7717/peerj.14110#supplemental-information.

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
