# Peer review of "Tubular epithelial progenitors are excreted in urine during recovery from severe acute kidney injury and are able to expand and differentiate in vitro"

_PeerJ, doi:10.7717/peerj.14110_

## Round 0.1 · original submission · Major Revisions

This manuscript is reporting interesting results on the characterization of kidney-derived progenitor cells present in AKI urine patients. However, in the present form, to support the validity of the findings, the authors should perform statistical analyses and indicate the number of samples/patients. After addressing the Reviewer's concerns, I will be able to understand if the manuscript deserves its publication.

Reviewer 1 ·

Basic reporting

The paper entiteled „Tubular epithelial progenitors are excreted in urine during recovery from severe acute kidney injury and are able to expand and differentiate in vitro” by Daniela Gerges et al., addresses the identification and characterization of kidney-derived progenitor cells present in the urine of patients with AKI

Starting with urine from patients with stage III AKI, morphological examination of urinary sediments and measurements of gene expression of nephron segment markers in samples of AKI with different etiologies were performed. Starting to characterize naïve epithelial cell clusters, expression of the markers Pax-8 and PROM1 was assessed by confocal immunofluorescence microscopy. Viable epithelial cells consisting of hyperplastic cell clusters were cultured and shown to proliferate for a minimum of 7 days. The cells in culture differentiated in renal cells expressing markers of specific segments.

The manuscript addresses an interesting topic, however, several serious concerns arise.
1. The major point of criticism is the insufficient delineation of the potential renal progenitors and their derivatives on one side and the urothelium-derived cells on the other side, particularly since the latter constitute the vast majority of cells in the urine.

2. Supposedly, not only urine-derived nephrospheres proliferate but also various cell types proliferate upon culturing, what is also associated with this kind of injury.

3. The RNA data in Fig.1 were based on a starting material of 800 ng RNA, but apparently not normalized to creatinine and therefore inconsistent results arise.

4. The performance of PCR form urine with the aim of quantification is problematic, as about 95% of cells are derived from urothelium and renal progenitors cover only a small amount of total urinary cells.

5. The AQP1 data are confounded by the fact that this aquaporine is also expressed in the urothelium.

6. This also holds true for Fig. 2 and in addition the Pax-8 staining of the putative renal epithelial clusters is not convincing.

7. In Fig. 3, the hyperplastic tubular cell clusters are not sufficiently characterized. And again, a precise exclusion from urothelium-derived cells is missing.

8. Finally the biological relevance of the data remains elusive.

Experimental design

s. above

Validity of the findings

see above

Reviewer 2 ·

Basic reporting

The manuscript by Gerges d et al. reports the isolation and amplification of progenitor cells from the urine of patients with AKI. The presence of these cells in the urine of AKI patient is a new observation and is potentially important. The manuscript is in general well written and clear with sufficient introduction, and relevant literature referenced. The authors characterize phenotypically and functionally the cells with good quality convincing images.

Experimental design

The research is original and meaningful. However, more details about the number of experiments and representation of quantitative assessment of the results shown, for example in a graph, with mean and standard error, particularly in Figure 5 and 6, would further improve the manuscript.

Validity of the findings

In conclusion, this is a manuscript reporting interesting and potentially important finding. The manuscript would benefit of adding quantitation of the data in graphs and details in Figure legend of the n of experiments. Also in Fig. 12 and 4 standard errors should be shown

·

Basic reporting

The paper with the title: “Tubular epithelial progenitors are excreted in urine during recovery from severe acute kidney injury and are able to expand and differentiate in vitro” aimed to address the issue of whether nephrosphere can be detected in non-transplanted patients with acute kidney injury (AKI) and whether its presence is also an indication of recovery in this patient cohort.

Experimental design

The principle of the study is clinically relevant, as it helps, with the aid of these tubular epithelial progenitors, to distinguish patients who might recover from stage 3 AKI from those who do not.

Validity of the findings

The entire paper is well written, the topic has been introduced in detail, the methods have been described in detail, and the results have been adequately presented and discussed.

Additional comments

However, I have following comments:

Abstract:
In this part, it is not clear whether this study was performed on human or animal subjects.

Materials and Methods - Patients:
Please determine number of the patients included in this study.

Results:
- I am curious to know if the age of the patients who enrolled in this study played a role in the production of this progenitor.

- To give the reader a better overview of the results of the study, I recommend creating a table showing the type and the day of these proginator cells production.

Conclusions:
To have such a conclusion that the mentioned tubule epithelial progenitor cells are positive indicators of renal recovery, the study must prove that patients who do not produce these progenitor cells do not recover from this stage of AKI. Please clearify.

---

## Round 0.2 · accepted · Accept

The authors have addressed reviewers' comments.

I am happy with the current version but there are some typos in the text (ex Baumann capsule instead Bowman capsule), as evidenced by the Reviewer. Please address these while in production

Reviewer 2 ·

Basic reporting

The authors have addressed all my concerns. Please check carefully cause there are some typos in the text (for ex Baumann capsule instead Bowman capsule).

Experimental design

No comment

Validity of the findings

No comment

Additional comments

No.comment

·

Basic reporting

This study aimed to address the issue of whether nephrosphere can be detected in non-transplanted patients with acute kidney injury (AKI) and whether its presence is also an indication of recovery in this patient cohort.

Experimental design

The principle of the study is clinically relevant, as it helps, with the aid of these tubular epithelial progenitors, to distinguish patients who might recover from stage 3 AKI from those who do not.

Validity of the findings

The entire paper is well written and the topic has been introduced in detail. In this revised version of the article, the authors have described the "Methods" part in more detail and presented and discussed the results in a more adequate way.

Additional comments

Thanks for responding to all my queries.